# Overexpression of Wild Soybean Expansin Gene *GsEXLB14* Enhanced the Tolerance of Transgenic Soybean Hairy Roots to Salt and Drought Stresses

**DOI:** 10.3390/plants13121656

**Published:** 2024-06-14

**Authors:** Linlin Wang, Tong Zhang, Cuiting Li, Changjun Zhou, Bing Liu, Yaokun Wu, Fumeng He, Yongqing Xu, Fenglan Li, Xu Feng

**Affiliations:** 1College of Life Sciences, Northeast Agricultural University, Harbin 150030, China; 18315407194@163.com (L.W.); 15147138029@139.com (T.Z.); licuiting123789@163.com (C.L.); hefumeng@neau.edu.cn (F.H.); doctor_smith@163.com (Y.X.); 2Daqing Branch of Heilongjiang Academy of Agricultural Sciences, Daqing 163316, China; andazhouchangjun@163.com (C.Z.); luibing528@163.com (B.L.); wuyaokun530@126.com (Y.W.); 3College of Agriculture, Northeast Agricultural University, Harbin 150030, China

**Keywords:** wild soybean, expansin, *GsEXLB14*, abiotic stress, hairy root

## Abstract

As a type of cell-wall-relaxing protein that is widely present in plants, expansins have been shown to actively participate in the regulation of plant growth and responses to environmental stress. Wild soybeans have long existed in the wild environment and possess abundant resistance gene resources, which hold significant value for the improvement of cultivated soybean germplasm. In our previous study, we found that the wild soybean expansin gene *GsEXLB14* is specifically transcribed in roots, and its transcription level significantly increases under salt and drought stress. To further identify the function of *GsEXLB14*, in this study, we cloned the CDS sequence of this gene. The transcription pattern of *GsEXLB14* in the roots of wild soybean under salt and drought stress was analyzed by qRT-PCR. Using an *Agrobacterium rhizogenes*-mediated genetic transformation, we obtained soybean hairy roots overexpressing *GsEXLB14*. Under 150 mM NaCl- and 100 mM mannitol-simulated drought stress, the relative growth values of the number, length, and weight of transgenic soybean hairy roots were significantly higher than those of the control group. We obtained the transcriptomes of transgenic and wild-type soybean hairy roots under normal growth conditions and under salt and drought stress through RNA sequencing. A transcriptomic analysis showed that the transcription of genes encoding expansins (EXPB family), peroxidase, H^+^-transporting ATPase, and other genes was significantly upregulated in transgenic hairy roots under salt stress. Under drought stress, the transcription of expansin (EXPB/LB family) genes increased in transgenic hairy roots. In addition, the transcription of genes encoding peroxidases, calcium/calmodulin-dependent protein kinases, and dehydration-responsive proteins increased significantly. The results of qRT-PCR also confirmed that the transcription pattern of the above genes was consistent with the transcriptome. The differences in the transcript levels of the above genes may be the potential reason for the strong tolerance of soybean hairy roots overexpressing the *GsEXLB14* gene under salt and drought stress. In conclusion, the expansin *GsEXLB14* can be used as a valuable candidate gene for the molecular breeding of soybeans.

## 1. Introduction

Soybean (*Glycine max*) is an economically important food and oilseed crop planted worldwide. It plays an important role in agricultural development as an important edible oil and feed crop [1,2]. However, the richness of the genetic variation in cultivated soybeans has gradually decreased during long-term artificial breeding and domestication. Salt, drought, low temperatures, and other environmental stresses usually lead to a reduction in soybean yield and quality, causing significant economic losses [3,4]. Wild soybeans (*Glycine soja*) are the ancestor and a related species to cultivated soybeans. To adapt to the harsh environmental conditions in the wild, wild soybean has gradually evolved disease resistance, insect resistance, stress resistance, and other excellent traits in its long-term evolution. Wild soybean is a precious plant genetic resource that is of great value for improving the resistance of cultivated soybean and enriching its genetic diversity [5,6].

Expansins are a class of non-enzymatically active cell-wall-relaxing proteins that are widely present in plants and were discovered and named in 1992 [7,8]. Expansins are usually divided into four subfamilies, of which the α-expansin subfamily (EXPA) was first discovered in the coleoptile of Avena sativa and the hypocotyl of Cucumis sativus, while β-expansin (EXPB) was identified in the pollen of gramineous plants. The γ-expansin (EXLA) and δ-expansin subfamily (EXLB) was identified after the completion of the whole-genome sequencing of rice (*Oryza sativa*) and *Arabidopsis thaliana* [9,10].

Among them, α-expansins rapidly induce the creep and stress relaxation of primary cell walls in a pH-dependent manner, carried out in a non-enzymatic manner [11]. β-expansins are usually highly specifically expressed in herbaceous plant pollen [12]. The EXLA and EXLB families are usually smaller in size and are believed to have originated later than the EXPA and EXPB families in terms of system evolution [11]. Although studies on expansin proteins have covered all subfamilies, most of the research has focused on the EXPA and EXPB families [10].

The changes in cell wall morphology mediated by expansins are crucial for plant growth, development, and response to external environmental stress [11,13]. The functions performed by the members of four subfamilies are generally similar. At present, many studies have proven that expansins are involved in regulating almost all processes of plant growth and development, such as seed germination [14], root development [15,16,17], nodule development [18,19], leaf growth [20], stomatal opening and closing [21], stem elongation [22], flower development [23], fruit ripening [24,25], seed yield [26,27], etc. In addition, expansins are involved in the response process of plants to various abiotic stresses, such as drought [28], salt [29], high temperature [30], low temperature [31], heavy metals [32], nutrient deficiency [33], oxidative stress [34], etc., and most of them show positive regulatory effects. Promoting growth and improving resistance are the main goals of crop molecular breeding, and expansins have become the focus of related research because of their functions.

The expansin gene family in wild soybeans may be of great significance for soybean molecular breeding, but related research is scarce. Our research group completed the whole-genome identification of the wild soybean expansin family in a previous study and analyzed the expression patterns of some expansin genes [35]. Previous studies have shown that *GsEXLB14* is specifically transcribed in wild soybean roots, and its transcript levels are significantly upregulated under salt and drought stress treatments. To further identify the function and mechanism of *GsEXLB14*, soybean hairy roots overexpressing this gene were obtained, and the phenotype and gene transcription profiles of transgenic hairy roots under salt and drought stress were determined. This study aims to provide a new candidate wild soybean expansin gene for molecular breeding work related to improving the environmental stress resistance of cultivated soybean.

## 2. Results

### 2.1. Basic Information of GsEXLB14 Amino Acid Sequence and Construction of Evolutionary Tree

We cloned the *GsEXLB14* gene using wild soybean roots as materials. The results of the analysis using the online software Expasy ProtParam tool (https://web.expasy.org/protparam/, accessed on 30 May 2022) showed that the GsEXLB14 protein is composed of 251 amino acids, with a molecular weight of 28,261.75, a protein weight of 28.27 kilodaltons, and a theoretical isoelectric point of 5.32; it is a stable protein. An amino acid phylogenetic tree analysis of GsEXLB14 and the expansin family members in soybean showed that this protein and GmEXLB3 were on the same evolutionary branch (Figure 1) and belonged to the EXLB subfamily.

### 2.2. Transcriptional Pattern of GsEXLB14 Gene under Salt and Drought Stress

We detected the transcription patterns of the *GsEXLB14* gene in the roots of wild soybean under salt- and mannitol-simulated drought stress by qRT-PCR. Under 150 mM NaCl stress, the transcription of *GsEXLB14* in the root system of wild soybean initially increased, and then decreased over time. At 8 h after treatment, the accumulation of transcripts of this gene was 2.73 times that before treatment, reached its peak, and then decreased significantly. Drought stress was simulated using 100 mM mannitol, and, after 4 h of treatment, the transcription of *GsEXLB14* showed a significant increase, reaching 3.47 times that of the untreated controls. However, the transcription levels decreased over time, similar to the pattern observed under salt stress (Figure 2). These results showed that salt and drought stress could promote the accumulation of *GsEXLB14* gene transcripts in the roots of wild soybean.

### 2.3. Observation on the Phenotype of Soybean Hairy Roots Overexpressing GsEXLB14 Gene

The soybean hairy roots overexpressing the *GsEXLB14* gene were obtained using the genetic transformation system mediated by *Agrobacterium rhizogenes* K599. We detected the transcription of the *GsEXLB14* gene in transgenic soybean hairy roots through RT-PCR experiments, and the results showed that *GsEXLB14* was successfully transcribed and that the plant materials could be used for further experiments (Figure 3B). The statistical result of the transgenic ratio of cotyledon nodes was 78% (n = 100). The results of the hairy root phenotype analysis are shown in Figure 4 and Figure 5. Compared to the K599 control group (soybean hairy roots induced by the empty bacteria control group), the overexpression of *GsEXLB14* significantly promoted the growth of transgenic hairy roots under normal growth conditions. The relative increases in the number of hairy roots, total root length, and total root weight were 2.07, 2.04, and 1.75 times that of the control group, respectively. Under 150 mM NaCl and 100 mM mannitol stress, the growth of both control and transgenic *GsEXLB14* soybean hairy roots was significantly inhibited, but the transgenic hairy roots showed a high tolerance. After 7 days of salt stress treatment, the average number of soybean hairy roots in the K599 control group increased by only 8.96, while the transgenic group increased by 21.34. Compared to the K599 control group, the total root length and root weight of transgenic hairy roots increased by 2.37 and 3.05 times, respectively. Under drought stress, the number of hairy roots in the control and transgenic groups increased by 14.54 and 26.91, respectively, after 7 d of treatment. The total length of hairy roots increased by 22.73 cm and 52.76 cm, and the total weight increased by 1.07 g and 3.38 g, respectively. These results showed that *GsEXLB14* overexpression significantly promoted the growth of soybean hairy roots under salt and drought stress.

### 2.4. Transcriptome Analysis of Soybean Hairy Roots Overexpressing GsEXLB14 Gene

We measured the transcriptomes of the K599 control and *GsEXLB14*-gene-overexpressing soybean hairy roots under normal growth conditions, salt stress, and drought stress, with three biological replicates for each treatment through RNA sequencing, for a total of 18 samples. A total of 202.38 Gb of Clean Data were obtained, with the Clean Data for each sample reaching 9 Gb, an error rate of 0.03%, a Q30 base percentage of ≥93%, and a GC base range of 43.22–44%. Sequence alignment was performed between the quality-controlled Clean Reads and the soybean reference genome, with an alignment rate of 96.31–97.12% for each sample, indicating that the sequencing data met the quality requirements. The number of differentially expressed genes between each group in the transcriptome is shown in Table 1, and a heatmap of the differentially expressed genes is plotted (Figure 6).

To preliminarily elucidate the effect of the overexpression of the *GsEXLB14* gene on the transcription of other genes in soybean hairy roots, we performed a functional annotation analysis of the differentially transcribed genes between each group. The KEGG classification information of differentially transcribed genes showed that, under normal conditions, the overexpressed *GsEXLB14* group had a higher number of differentially expressed genes in the Plant MAPK signaling pathway, plant hormone signal transduction, biosynthesis of secondary metabolites, metabolic pathways, and plant pathogen interaction categories than the control group (Figure 7). This suggests that *GsEXLB14* is involved in the regulation of plant abiotic and biotic stress responses and metabolic processes.

In addition, we analyzed the function and quantity of differentially expressed genes between different groups involved in plant environmental stress responses and abiotic stress resistance regulation under salt and drought stress, as shown in Figure 8. Under salt stress, the transcription of 13 types of genes was significantly upregulated in the hairy roots induced by K599 empty bacteria (control group), with the largest number of peroxidase genes (four), encoding glutathione S-transferases and sulfate transporters (three and two genes, respectively), including the desiccation-protective protein, superoxide dismutase, anion channel protein, MYB, and NAC transcription factors (Figure 8A). In the *GsEXLB14* overexpression group, 10 genes were significantly upregulated, with the highest number of sulfate transporters (four genes) and the upregulated transcription of sugar transporters, peroxidases, EXPB-type expansin proteins, and MYB transcription factors (Figure 8B). Under salt stress, the transcription of 15 types of genes in the transgenic group was significantly upregulated compared to the control group, mainly including: EXPB-type expansin, peroxidase, auxin-responsive protein, calcium/calmodulin-dependent protein kinase, and MYB transcription factor genes. In addition, it was also found that the transcription of genes that encode anion channel protein, vacuolar membrane proton pump, and H^+^-transporting ATPase in soybean hairy roots overexpressing *GsEXLB14* also significantly increased (Figure 8C).

Under drought stress simulated by 100 mM mannitol, 15 classes of resistance-related genes, including peroxidases, auxin transporters, sugar efflux transporters, and MYB transcription factors, were upregulated in the control group. The transcription of two members of the expansin protein family was also significantly upregulated (Figure 8D). A total of 11 categories of genes were significantly upregulated in the GsEXLB14 overexpression group, with a relatively small number of related genes. Among them, the number of genes encoding miraculin-like protein is the largest, with eight genes. Second, the upregulated transcription numbers of peroxidase and EXLB expansin genes were also relatively high, with four in each (Figure 8E). Compared to the control group under drought stress, 11 genes were significantly upregulated in transgenic hairy roots, among which peroxidase, calcium/calmodulin-dependent protein kinase, and pathogenesis-related protein 1 (PR1) had the highest number of genes. The second largest group was the expansin B/LB family members and AP2/ERF transcription factors, with two in each. It also included the dehydration-responsive protein RD22 and the sugar transport protein (Figure 8F).

To verify the accuracy of the changes in the transcription levels of differentially transcribed genes in the transcriptome, we separately selected five differentially expressed genes in the control and transgenic groups under salt and drought stress conditions and detected their transcription levels using qRT-PCR. Under salt stress, the five selected differentially expressed genes were expansin, peroxidase, H^+^ transport ATPase, the anion channel protein, and the vacuolar membrane ion pump. Among them, the transcript level of expansin (GLYMA_05G065300) in the overexpression *GsEXLB14* group was 550.51 times higher than that in the control group—a significant level. The transcript levels of the other four genes in the treatment group were also significantly higher than those in the control group (Figure 9A). The differentially expressed genes under drought stress were the expansin protein, peroxidase, AP2/ERF transcription factor, dehydration-responsive protein, and WRKY transcription factor. The transcription levels of the five genes in the treatment group were significantly higher than those in the control group, with the transcription of the peroxidase and expansin protein being 10.60 and 9.78 times higher than those in the control, respectively. The other three genes were significantly upregulated (Figure 9B). The qRT-PCR results were consistent with the transcriptome sequencing results.

## 3. Discussion

### 3.1. Transcription Patterns of GsEXLB14

The transcription pattern of the expansin gene was highly specific to different plant organs and was determined by its functions. For example, the *SlExp1* gene, which regulates the breakdown of tomato (*Solanum lycopersicum*) cell walls and fruit softening, is predominantly transcribed in fruit [36]. *SgEXPB1* is upregulated in *Stylosanthes guianensias* stylo roots under low phosphorus conditions and has been shown to be involved in root growth regulation under phosphorus-deficient conditions [37]. In our previous study, we found that the *GsEXLB14* gene in wild soybeans was specifically transcribed in the roots through transcriptome sequencing. This is similar to *BdEXPA27* in *Brachypodium distachyon*, *OsEXPB5* in *Oryza sativa*, and *HvEXPB7* in *Hordeum vulgare*, all of which have been shown to regulate root development [38,39,40]. Therefore, *GsEXLB14* may play a key role in wild soybean root growth.

In addition, we previously found that the transcription levels of *GsEXLB14* significantly increased after 4 h of treatment with 150 mM NaCl and 12 d of water deprivation. To further confirm the response of this gene to salt and drought stress, in this study, we treated wild soybean plants with salt and drought stress using NaCl and mannitol, respectively, and monitored the transcription level of *GsEXLB14* for 24 h. We found that, under both types of stress, the increase in the transcript levels of this gene was concentrated between 4 and 12 h after treatment (Figure 2). The root is the organ through which plants directly respond to water stress in the soil [41]. *GsEXLB14* is specifically transcribed in roots and responds actively to salt and drought stress. These data suggest that this gene plays an important role in regulating salt and drought tolerance in wild soybeans.

### 3.2. Potential Functions of GsEXLB14

Because the transcription of *GsEXLB14* is root-specific, we used the soybean hairy root transformation system to preliminarily identify the function of this gene. An amino acid phylogenetic tree analysis showed that the homologous protein of GsEXLB14 in soybean was GmEXLB3. However, there are few reports on the role of EXPB/LB-type expansin proteins in regulating plant resistance to environmental stresses (such as salinity, water, and temperature) in soybeans, with most studies focusing on the utilization efficiency of phosphorus in the soil and the growth of the root nodule [42,43]. In addition, *GmEXPA1* gene overexpression reduces plant susceptibility to *Meloidogyne incognita* [44]. In this study, the phenotypic observation results showed that *GsEXLB14* overexpression significantly promoted the growth of soybean hairy roots and improved their tolerance to salt and drought stress simulated by mannitol (Figure 4). The quantitative phenotypic data further confirmed this hypothesis (Figure 5).

The overexpression of expansins can promote plant root growth, as has been confirmed in various plants [45]. The first root-specific expression of the expansin protein GmEXP1, discovered in soybean, has been shown to play an important role in regulating root development, especially in the elongation of the main root and the initiation of lateral roots [46]. Although the mechanism by which expansin exerts its function has not yet been confirmed, it relaxes the cell wall to promote cell elongation. In the “acid growth” model, the pH-dependent increase in cell growth and wall extensibility occurs under several circumstances. During the auxin-mediated cell elongation, auxins activate a proton pump across the plasma membrane, which lowers the extracellular pH to activate expansin. The expansin protein then relaxes the cell wall without altering the covalent structure of the cell wall by disrupting the hydrogen bonds between the matrix polysaccharides and cellulose microfibrils. The relaxed cell wall, driven by turgor pressure, increases the absorption of water by protoplasts or vacuoles, thus increasing the cell volume [47,48,49]. This may explain why the overexpression of *GsEXLB14* significantly promoted the growth of soybean hairy roots. The auxin required in this process may originate from the cotyledons or vigorously developing root tips, which can promote cell elongation and further cell division, resulting in more vigorous growth than the control group. Transcriptome data showed that *GsEXLB14* overexpression altered the transcription of genes related to the MAPK cascade pathway, plant hormone signal transduction, and secondary metabolism processes. Some of these key genes may be potential factors promoting root growth, but this requires further research. A well-developed root system helps plants better resist osmotic and water stress, such as salt and drought, which also indicates the potential application value of *GsEXLB14* for the molecular breeding of resistant soybeans in cultivation.

Salt and drought stress are major abiotic factors that reduce soybean yield and restrict the development of the soybean industry. Improving the tolerance of cultivated soybeans to environmental stress is the key to solving this problem. Expansins have been shown to play a positive role in this regard. For example, in terms of regulating salt stress tolerance, *NtEXPA4* in tobacco (*Nicotiana tabacum*), *TaEXPA2* in wheat, *RhEXPA4* in rose (*Rosa hybrida*), and *OsEXPA7* in rice can improve salt tolerance in transgenic plants [38,50,51,52]. These genes function by increasing cell wall extensibility, reducing water loss, enhancing the activity of antioxidant enzymes and the content of osmotic regulatory substances, and regulating Na^+^/K^+^ ion accumulation. In terms of regulating plant drought stress tolerance, the overexpression of the expansin-like gene *GhEXLB2* in upland cotton (*Gossypium hirsutum*) significantly enhances drought resistance [53]. The *Arabidopsis* expansin gene *AtEXPA18* ameliorates drought stress tolerance in transgenic tobacco plants [54]. The *Brassica rapa* expansin-like B1 gene, *BrEXLB1*, is involved in regulating root development and drought stress responses [55]. When plants are subjected to salt and drought stress, the upregulation of expansin gene transcription in the root system due to water deficiency can be considered a stress response. This response promotes root growth, increases the root–shoot ratio, and enhances water absorption from the soil, thereby improving the resistance. This may also be a potential reason for the superior growth status of hairy roots overexpressing the *GsEXLB14* gene under NaCl and mannitol stress compared to the control group.

To further reveal the transcription of genes in transgenic soybean hairy roots under salt and drought stress and to elucidate the functional mechanism of the *GsEXLB14* gene from the perspective of gene transcription, we determined the transcriptome of hairy roots under different treatments and obtained differentially expressed genes related to plant environmental stress resistance regulation (Figure 8). Under salt stress, *GsEXLB14* overexpression significantly promoted the transcription of EXPB-like expansin genes in cultured soybean hairy roots, and the transcription of auxin response protein genes also increased significantly, which is consistent with the theory of “acid growth” and will better promote the growth of roots. In addition, the transcription of some protein kinases and transcription factors was significantly upregulated, and these genes were all related to the regulation of plant salt stress resistance. The upregulation of genes closely related to ion transport, such as anion channel proteins, vacuolar membrane ion pumps, and H^+^-ATPases, plays a key role in alleviating the ion toxicity damage caused by salt stress. Interestingly, under drought stress, the number of upregulated miraculin transcripts in the transgenic hairy roots was the highest (Figure 8E). This protein is a glycoprotein that primarily exists in plants of the family Sapindaceae, including the mysterious fruit (*Synsepalum dulcificum*), and its main function is to improve taste [56]. There have been few reports on this family of proteins in legumes. Compared to the control group, the significantly upregulated genes in the transgenic group under drought stress were closely related to the regulation of plant responses to drought stress and the alleviation of stress-induced cell damage, including peroxidase, expansin, and dehydration response proteins. The upregulation of these genes may explain why transgenic soybean hairy roots grew better under drought stress than the control group. Although the function and mechanism of *GsEXLB14* have not yet been systematically revealed, preliminary research results indicate its potential value in improving the tolerance of cultivated soybeans to salt and drought stress.

## 4. Materials and Methods

### 4.1. Cloning of GsEXLB14 Gene

Wild soybean seeds (W05) were provided by the College of Life Sciences of Northeast Agricultural University. Uniformly sized plump seeds were treated with 98% H_2_SO_4_ for 10 min to break dormancy, washed three times with sterile water, and planted in sterile Petri dishes with double-layer filter paper. Sterile water was used to keep the filter paper moist and at 25 °C for dark cultivation. When the radicle broke through the seed coat by about 1 cm, the germinated seeds were transplanted into a nutrient bowl (9 cm diameter, 100% vermiculite + Hoagland nutrient solution) for culture (25/22 °C day and night temperature, 16/8 h light cycle). After 30 d, wild soybean plants with the same growth were selected, and the roots was sampled (0.2 g), and then stored at −80 °C after quick freezing in liquid nitrogen.

Easypure^®^ Plant RNA Kit (TransGen Biotechnology Co., Ltd., Beijing, China) was used to extract total RNA from the root samples of the above wild soybean plants. RNA was reverse-transcribed with TransScript^®^ IV One-Step gDNA Removal and cDNA Synthesis SuperMix (TransGen Biotechnology Co., Ltd., Beijing, China) to obtain cDNA. The wild soybean cDNA obtained in the above steps was used as the template, and the GsEXLB14-Clone-FW and GsEXLB14-Clone-RV primers and EasyTaq^®^ DNA Polymerase (TransGen Biotechnology Co., Ltd., Beijing, China) kit were used for PCR amplification to obtain the amplification product containing *GsEXLB14* gene. The primers used in this study are listed in Appendix A. PCR amplification products were detected using 1.2% agarose gel electrophoresis, and the target size fragments were purified and recovered using EasyPure^®^ PCR Purification Kit (TransGen Biotechnology Co., Ltd., Beijing, China). The recovered product was ligated with T3 cloning vector using pEASY^®^-T3 Cloning Kit to obtain recombinant plasmid, which was named pEASY-T3-GsEXLB14. The constructed plasmid was transformed into competent *E. coli* (Trans1-T1 phage-resistant chemically competent cell, TransGen Biotechnology Co., Ltd., Beijing, China) and sent to a sequencing company for sequencing. The amino acid sequence of GsEXLB14 was analyzed using online software ExPASY ProtParam tool (https://web.expasy.org/protparam/, accessed on 30 May 2022). The amino acid evolutionary tree of GsEXLB14 and members of the soybean expansin family were constructed using MEGA 7.0, and the amino acid sequence of the soybean expansin family was obtained from Zhu et al. 2014 [57].

### 4.2. Transcriptional Pattern Analysis of GsEXLB14 Gene under Salt and Drought Stress

The planting mode of wild soybeans was the same as above, and the germinated seeds were treated after being transplanted into a nutrient bowl for 30 d. Salt and drought stress treatments were performed using 150 mM and 100 mM mannitol, respectively. Each nutrient bowl was irrigated with 50 mL of rhizosphere soil, and the control group was irrigated with an equal volume of sterile water. Roots were sampled at 0, 4, 8, 12, and 24 h after treatment. The transcript level of *GsEXLB14* was detected via qRT-PCR using the TRANSGEN Top Green qPCR SuperMix kit (TransGen Biotechnology Co., Ltd., Beijing, China). The amplification of *Actin-11* (GenBank: LOC114395252) in wild soybeans was used as an internal control. The expression levels for *GsEXLB14* were determined using the 2^−∆∆CT^ method, and relative transcript levels were calculated and normalized as described previously [58].

### 4.3. Construction of Overexpression Vector of GsEXLB14 Gene

The *GsEXLB14* gene was overexpressed using CaMV35S promoter (Figure 3A). Using the pEASY-T3-GsEXLB14 cloning vector as the template, the GsEXLB14-*Sma*I-FW and GsEXLB14-*Sma*I-RV primers and EasyTaq^®^ DNA Polymerase kit (TransGen Biotechnology Co., Ltd., Beijing, China) were used to add *Sma*I restriction sites to the upstream and downstream of the GsEXLB14 gene via PCR. The PCR products were purified and ligated into a T3 cloning vector for sequencing. The resulting vector was named pEASY-T3-GsEXLB14-*Sma*I. The pEASY-T3-GsEXLB14-*Sma*I vector was digested with the restriction enzyme *Sma*I (New England Biolabs Ltd., Beijing, China), and a small fragment (759 bp) obtained from the digestion product was purified for future use. The pCAMBIA1302 expression vector was digested using the restriction enzyme *Pml*I (New England Biolabs Ltd., Beijing, China). The digested product was dephosphorylated using calf intestinal alkaline phosphatase (CIP, New England Biolabs Ltd., Beijing, China) and purified. The purified target digested products of pEASY-T3-GsEXLB14-*Sma*I and the pCAMBIA1302 vector were ligated using T4 DNA ligase (New England Biolabs Ltd., Beijing, China) to obtain the recombinant expression vector pCAMBIA1302-GsEXLB14. The recombinant expression vector was identified using primers GsEXLB14-identify-FW and GsEXLB14-identify-RV.

### 4.4. GsEXLB14 Overexpression through Soybean Hairy Roots

The recombinant expression vector, pCAMBIA1302-GsEXLB14, was transformed into *Agrobacterium rhizogenes* K599 (Weidi Biotechnology Co., Ltd., Shanghai, China) via freeze–thawing. The transformation of cultivated soybean hairy roots was performed as described by Li et al. 2014 [59]. The soybean variety cultivated was Dongnong 50, and the seeds were provided by the College of Agriculture of the Northeast Agricultural University. The medium for the co-culture of soybean cotyledon nodes with *Agrobacterium rhizogenes* was 1/10 MS solid medium (sucrose as the carbon source, 30 g/L; agarose, 7 g/L), and the medium for hairy root induction was 1/2 MS solid medium (sucrose as the carbon source, 30 g/L; agarose, 7 g/L). The induction condition was 28 °C, and the light cycle was 16/8 h. The soybean hairy roots induced by K599 empty bacteria were used as a non-transgenic control group. RT-PCR was used to identify the transcription of *GsEXLB14* gene in hairy roots. RNA extraction and cDNA synthesis were performed as previously described. Soybean *Actin* (GenBank: LOC100798052) was used as an internal reference gene.

### 4.5. Phenotypic Observation on Hairy Roots of Soybean Overexpressing GsEXLB14 Gene

Transgenic hairy roots and K599 null bacteria-induced hairy roots (control group) were selected as the experimental materials. Single hairy roots were transferred to 1/2 MS solid medium or 1/2 MS solid medium containing 150 mM NaCl (salt stress) or 100 mM mannitol (drought stress). After 7 d, the number, total root length, and root weight of hairy roots were counted, and the relative growth of the number, total root length, and total root weight of hairy roots were calculated; photographs were taken at the same time.

### 4.6. Transcriptome Assay of Hairy Roots

The hairy roots of soybeans induced by K599 null strain and overexpressing *GsEXLB14* gene under normal growth conditions, salt stress, and drought stress were sampled. The samples were snap-frozen in liquid nitrogen and sent to Metware Biotechnology Inc. (Wuhan, China) on dry ice for transcriptome sequencing. The fragment sizes and concentrations of the libraries were determined using an Agilent 2100 Bioanalyzer. The library was sequenced using the Illumina HiSeq platform. CuffQuant and CuffNorm use fragments per kilobase of transcript per million fragments mapped (FPKM) as indicators of transcript or gene expression levels. For the screening of differentially transcribed genes among groups, FPKM ≥ 5 and transcription upregulation multiple ≥5 were used as thresholds. The differentially transcribed genes were functionally annotated through NCBI and KEGG databases. Genes involved in the regulation of plant environmental stress responses, and abiotic stress resistance was screened and counted. Simultaneously, a total of 10 significantly differentially transcribed genes in the control group and transgenic group under salt and drought stress were randomly selected for qRT-PCR identification; methods were the same as above.

### 4.7. Statistical Analysis

All trials were repeated at least three times, and the data are presented as the mean ± standard deviation. GraphPad Prism 8 software was used for statistical analysis and plotting.

## 5. Conclusions

In the current study, we cloned the expansin gene *GsEXLB14* from the roots of wild soybean. The transcription levels of *GsEXLB14* in the roots of wild soybean significantly increased under salt- and mannitol-simulated drought stress. The overexpressing *GsEXLB14* gene significantly promoted the growth of cultivated soybean hairy roots and their tolerance to salt- and mannitol-simulated drought stress. The data from the transcriptome analysis showed that, under salt- and mannitol-simulated drought stress, the transcripts encoding expansins, peroxidases, H^+^-ATPases, calcium/calmodulin-dependent protein kinases, and dehydration-response proteins in transgenic soybean hairy roots showed significant accumulation, which may be the reason why transgenic hairy roots grow better under stress.

## Figures and Tables

**Figure 1 plants-13-01656-f001:**
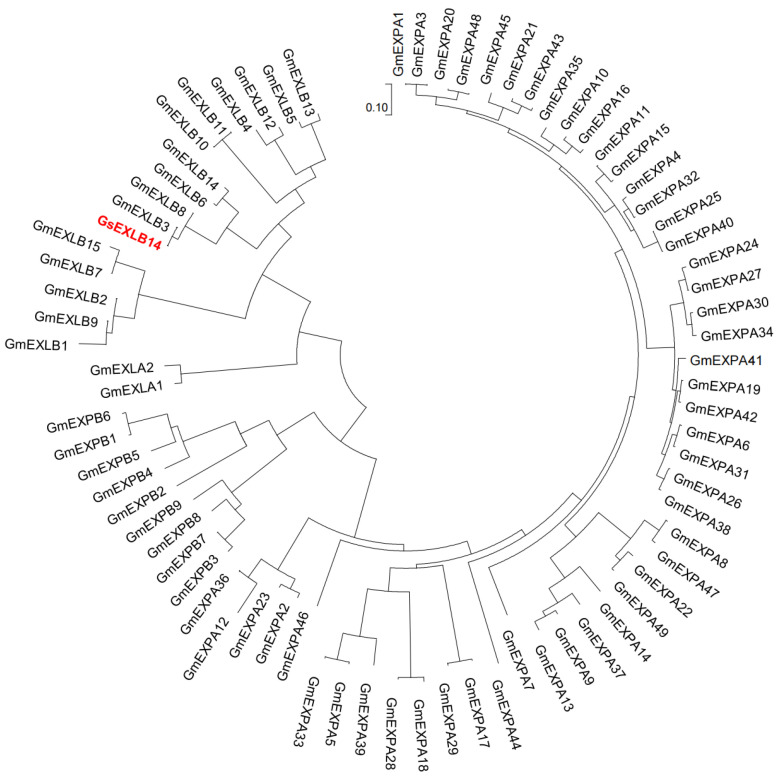
Amino acid phylogenetic tree of GsEXLB14 and soybean expansin family members.

**Figure 2 plants-13-01656-f002:**
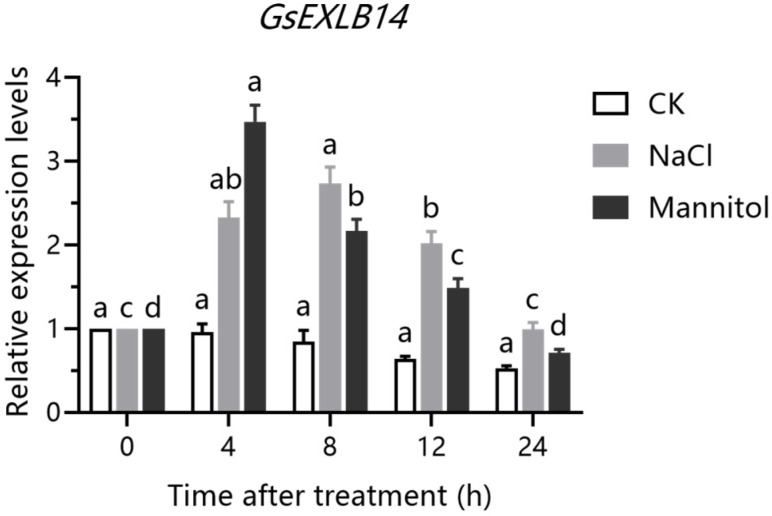
Transcriptional patterns of *GsEXLB14* in wild soybean roots under salt and drought stresses. Note: The experiment was conducted using a relative quantitative method, and the relative expression level represents the fold in the transcription level of *GsEXLB14* gene relative to that at 0 h after treatment. Different lowercase letters indicate that the transcript level of *GsEXLB14* under same treatment condition was significantly different between different treatment times (*p* < 0.05).

**Figure 3 plants-13-01656-f003:**
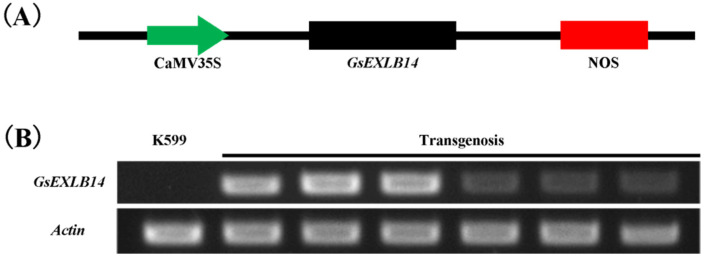
Construction of overexpression vector of *GsEXLB14* and RT-PCR detection results of transgenic soybean hairy roots. Note: (**A**) Schematic diagram of overexpression vector construction. (**B**) RT-PCR detection results of transgenic soybean hairy roots. K599 represents the soybean hairy roots induced by the empty bacteria (without any exogenous expression vector).

**Figure 4 plants-13-01656-f004:**
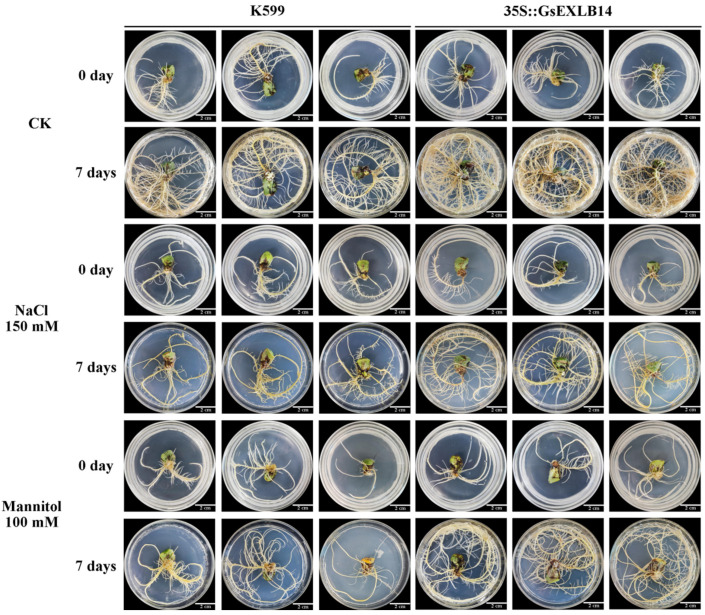
Phenotype of soybean hairy roots overexpressing *GsEXLB14* under normal, salt, and drought stress conditions (Bar = 2 cm).

**Figure 5 plants-13-01656-f005:**
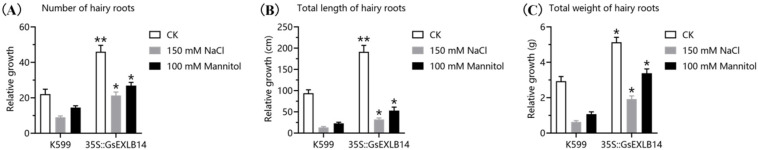
Statistical results of growth indicators of soybean hairy roots overexpressing *GsEXLB14* gene under salt and drought stress. Note: Relative growth indicates the change in different phenotypes of soybean hairy roots after 7 d under different treatments relative to the original state. (**A**) Relative increase in the number of hairy roots. (**B**) Relative growth of total length of hairy roots. (**C**) Relative increase in total weight of hairy roots. (* indicates that, under the same treatment condition, there was a significant difference between transgenic group and K599 control group, * *p* < 0.05, ** *p* < 0.01, n = 30.)

**Figure 6 plants-13-01656-f006:**
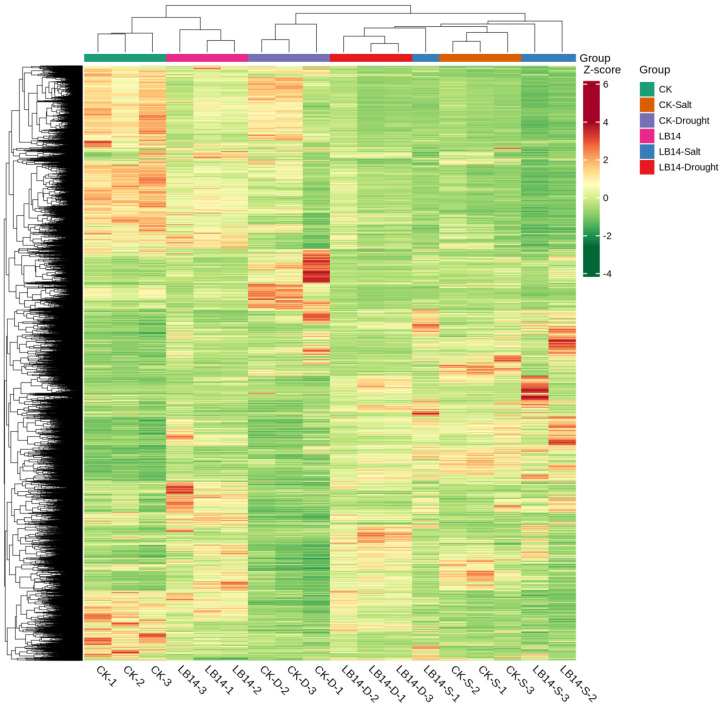
Clustering heatmap of differential genes among treatment groups in the transcriptome.

**Figure 7 plants-13-01656-f007:**
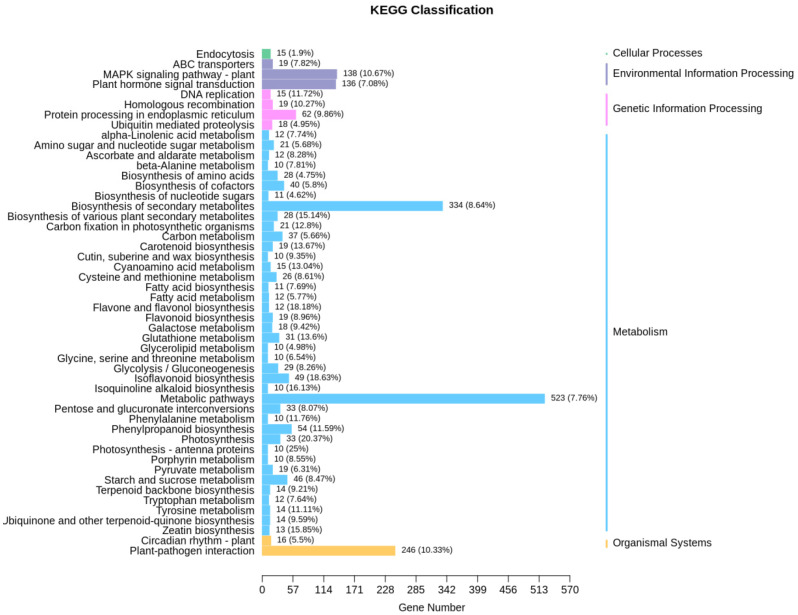
KEGG classification information of differentially transcribed genes in overexpressing *GsEXLB14* and control group.

**Figure 8 plants-13-01656-f008:**
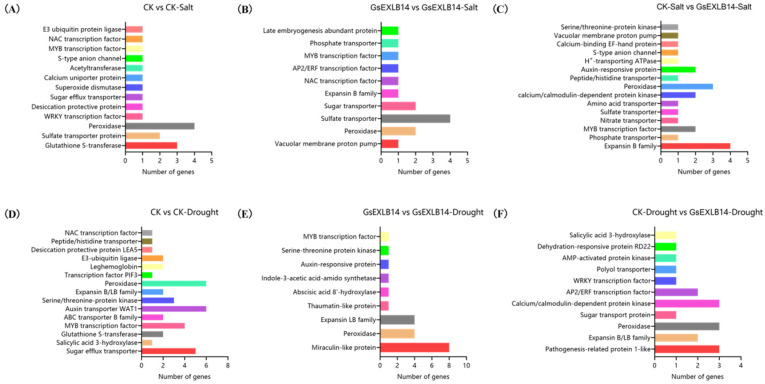
Number of upregulated transcription genes involved in plant environmental stress response and abiotic stress resistance regulation between the treatments groups (left group of “vs” was used as control). Note: Data were derived from the transcriptome, with FPKM value ≥ 5 and transcriptional upregulation fold ≥5 as thresholds. (**A**) CK vs. CK-Salt. (**B**) GsEXLB14 vs. GsEXLB14-Salt. (**C**) CK-Salt vs. GsEXLB14-Salt. (**D**) CK vs. CK-Drought. (**E**) GsEXLB14 vs. GsEXLB14-Drought. (**F**) CK-Drought vs. GsEXLB14-Drought.

**Figure 9 plants-13-01656-f009:**
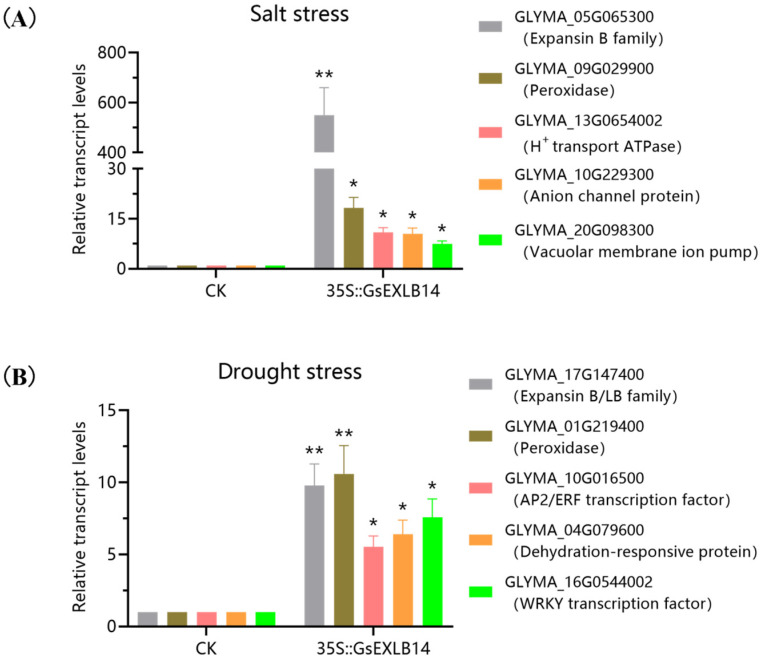
qRT-PCR results of differentially transcribed genes between control group and overexpressing *GsEXLB14* group under salt and drought stress. Note: The experiment was conducted using a relative quantitative method, and the relative expression level represents the fold in the transcription level of gene relative to CK. (**A**) Salt stress. (B) Drought stress. (* indicates that, under the same treatment, the transcriptional level of candidate gene was significantly different between overexpression group and control group, * *p* < 0.05, ** *p* < 0.01.)

**Table 1 plants-13-01656-t001:** Number of differentially transcribed genes between groups in transcriptome.

Group	Total	Down	Up
CK_vs_CK-Drought	4496	2836	1660
CK_vs_CK-Salt	7314	4054	3260
CK_vs_LB14	3314	827	2487
LB14_vs_LB14-Drought	2412	1551	861
LB14_vs_LB14-Salt	4434	3307	1127
CK-Drought_vs_LB14-Drought	6551	2790	3761
CK-Salt_vs_LB14-Salt	3580	2231	1349

## Data Availability

Data are contained within the article and Appendix A.

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
