# Peer review of "Overexpression of Wild Soybean Expansin Gene *GsEXLB14* Enhanced the Tolerance of Transgenic Soybean Hairy Roots to Salt and Drought Stresses"

_plants, 2024, doi:10.3390/plants13121656_

Round 1

Reviewer 1 Report

Comments and Suggestions for Authors

Comments to the authors:

I enclosed my comments to the Manuscript entitled “Overexpression of wild soybean expansin gene GsEXLB14 enhanced the tolerance of transgenic soybean hairy roots to salt and drought stresses” by Wang” et al.

The manuscript presents promising data on the effect of overexpressing gene GsEXLB14 from wild soybean on Glycine max root growth and development and in the response to abiotic stresses (drought and salinity) at the physiological and transcriptomic levels. Nevertheless, beyond needing an important English editing, the manuscript contains numerous formatting and content errors that must be corrected so that the work can be evaluated from a scientific point of view.

I explain below some general and specific issues that needs to be addressed. I just mention some examples, but there are more:

1- Abstract: it does not clearly indicate some aspects of the methodological approach used to achieve the results presented;

The expressions  ”high quality resistance genes”, “significantly better” what does mean?; it is not clear if the GsEXLB14 gene has been identified here; reading further I understood that it corresponds to previous works..; It is not clearly stated that the transcriptomic analyses come from an RNA seq experiment and that they have validated some of these genes by QRT PCR. It is not indicated either in the abstract or clearly in the Results section; It is only mentioned in the materials and methods section.

2- Introduction:

The introduction does not contain the necessary information on the topic and the state of the art of the subject, and it's disorganized.

the expressions “main economic crops”(37) is not correct;

The way of writing is repetitive and makes it not very fluid throughout the Ms(55-56; 57-58);

“Seventy-five members of the expansin family have been identified with a total of 75 members” (63-64): this sentence has no sense, and gives an idea that the manuscript is in draft phase;

“provide new high-quality genetic resources” (83): what does it mean?

In this section but also in the discussion: the authors do not indicate to the criteria for classifying extensins into different groups: alpha, beta, gamma... the protein under study belongs to the gamma type, but  it is not clear: each group has been recruited for different responses/functions? Based on? The authors mentioned several examples for the reported role of expansins belonging to different groups in response to different abiotic and biotic stresses in different plant species “jumping” from one to another. I assume that the authors want to emphasize the importance of this family of proteins in these processes, but in the way it is presented, it does not meet that objective, I do not follow the rationale. On the contrary, it is confusing, it seems as if there were no script.

3- Results:

As a general comment: although methodological details should be included in the corresponding materials and methods section, it is essential that the results section includes relevant information on how these results were obtained, the biological material, etc. The results section in the current Ms present data, sample nomenclature, treatments, etc., without explanation of their origin. Furthermore, many of this information does not appear in the materials and methods section either.

“is a stable protein”(91): why?

Figures and table: this is really important, any of the figure legends are self-explanatory. On the contrary, they do not explain the nomenclature used, what the relative expression values refer to, etc. In addition, the format of the figure legends is incorrect. In addition, some sections of the figures are not mentioned in the text. This is the case of part A of figure 3.

There is a lot of relevant information missing, such as that concerning the biological material under study, the treatments, the replicates, some of the procedures…. This information is crucial to follow this work.

“These results suggest that salt and drought stress can induce positive transcription of GsEXLB14 in the roots of wild soybeans.”(106-107): I do not understand: Why “suggest”: the authors shows induction?

(114)The authors have already presented RT PCR data in the previous section, although they do not explain it, but now they mention it. Is it assumed to be end-time PCR? (114). After the epigraph, the paragraph begins without explaining that they are referring to the roots that overexpress the gene under study. As mentioned before, k599, control... what does this nomenclature correspond to?

There is no allusion or explanation that the result of transformation is roots that overexpress the protein under study: is it uniform throughout the root? Is it stable? . genotyping?

In relation to the gene expression results, beyond the legends of the missing figures: reference gene used, replicates, number of times the experiment has been repeated, are the primers specific? Is has been proven that they are not amplifying the endogenous G. max gene? In this sense I have many doubts. The expression values ​​for K 599 are considered to be 1... but what is exactly this sample “control K599”?

(117) “statistical result ”: what does it mean?

(121) “normal growth conditions ”: what does it mean?

(125) is poorly expressed.

(146): Figure 5. Phenotypic quantification results of soybean: is poorly expressed.

(149): “separately”: what does men in this context?

(152): it is assumed that these results come from RNA seq experiment. It is not indicated. Then the validation of some of this genes by q real time : which criteria was used to select those genes from the whole RNS seq data?

There are several incorrect or poorly expressed sentences or expressions: 166,181,191,202, 218, 219…..

3- Discussion:

The discussion is disorganized and, for all the reasons mentioned, it is difficult to follow throughout it. As mentioned before, in this sections there are as well incorrect or poorly expressed sentences. There is barely discussion of the data.

4- Materials and methods: I already comment on that:

A first epigraph that is missing for Plant material, treatments and experimental design and sampling: I consider that is crucial a clear description of the plant material used, plant starting material for each experiment, biological, technical replicates, and indicate each replicate consists in how many samples, material,..?

Many necessary information is missing, such as that concerning the biological material under study, the treatments, the replicates carried out, some of the procedures described, such as PCR conditions, biomolecule extraction protocols, among others.

Concerning the A. rhyzogenes transformation: an explanation about the resulting transgenic material: transient or stable root transformation….It is important to know the nature of the plant material used in  the experiments

Author Response

Dear Editor and Reviewer 1:

Thank you for your letter and for the reviewer’ comments concerning our manuscript entitled “Overexpression of wild soybean expansin gene GsEXLB14 en-hanced the tolerance of transgenic soybean hairy roots to salt and drought stresses” (Manuscript ID: plants-3022279). Those comments are all valuable and very helpful for revising and improving our paper, as well as the important guiding significance to our researches. We have studied comments carefully and have made correction which we hope meet with approval. Revised portion are marked in red in the paper. The main corrections in the paper and the responds to the reviewer’s comments are as flowing:

Responds to the reviewer’s comments:

Reviewer #1:

  1. Response to comment: (1- Abstract: it does not clearly indicate some aspects of the methodological approach used to achieve the results presented;

The expressions  ”high quality resistance genes”, “significantly better” what does mean?; it is not clear if the GsEXLB14 gene has been identified here; reading further I understood that it corresponds to previous works..; It is not clearly stated that the transcriptomic analyses come from an RNA seq experiment and that they have validated some of these genes by QRT PCR. It is not indicated either in the abstract or clearly in the Results section; It is only mentioned in the materials and methods section.)

Response: First of all, thank you very much for your valuable comments. Indeed, in the current manuscript, the abstract does not fully summarize the content of the article. We have revised the summary according to your comments.

(1) “high quality resistance genes”

The original statement was prone to misinterpretation, and we have made the necessary changes. The revised statement is: Wild soybeans have long existed in the wild environment and possess abundant resistance gene resources, which hold significant value for the improvement of cultivated soybean germplasm. In our previous study, we found that the wild soybean expansin gene GsEXLB14 is specifically transcribed in roots, and its transcription level significantly increases under salt and drought stress. To further identify the function of GsEXLB14, in this study, we cloned the CDS sequence of this gene. The transcription patterns of GsEXLB14 in the roots of wild soybean under salt and drought stress was analyzed by qRT-PCR. This change helps readers better distinguish between previous and current work.

(2) “significantly better” what does mean?

This sentence means that under salt and drought stress, the growth status of soybean hairy roots overexpressing GsEXLB14 gene is better than that of wild type. The description here is indeed not rigorous enough, and we have made changes. The revised statement is: “Under 150 mM NaCl and 100 mM mannitol-simulated drought stress, the relative growth values of the number, length, and weight of transgenic soybean hairy roots were significantly higher than those of the control group.

(3) It is not clearly stated that the transcriptomic analyses come from an RNA seq experiment and that they have validated some of these genes by QRT PCR. It is not indicated either in the abstract or clearly in the Results section; It is only mentioned in the materials and methods section.

We have added a description of transcriptome sequencing in the abstract, as per your suggestion, and also included the identification of the transcription levels of certain genes through qRT-PCR. The revised statement is: “We obtained the transcriptomes of transgenic and wild-type soybean hairy roots under normal growth conditions and under salt and drought stress through RNA sequencing.” At the same time, the following was added to the abstract: “The results of qRT-PCR also confirmed that the transcription pattern of above genes was consistent with the transcriptome.

  1. Response to comment: “2- Introduction:

The introduction does not contain the necessary information on the topic and the state of the art of the subject, and it's disorganized.

the expressions “main economic crops”(37) is not correct;

The way of writing is repetitive and makes it not very fluid throughout the Ms(55-56; 57-58);

“Seventy-five members of the expansin family have been identified with a total of 75 members” (63-64): this sentence has no sense, and gives an idea that the manuscript is in draft phase;

“provide new high-quality genetic resources” (83): what does it mean?

In this section but also in the discussion: the authors do not indicate to the criteria for classifying extensins into different groups: alpha, beta, gamma... the protein under study belongs to the gamma type, but  it is not clear: each group has been recruited for different responses/functions? Based on? The authors mentioned several examples for the reported role of expansins belonging to different groups in response to different abiotic and biotic stresses in different plant species “jumping” from one to another. I assume that the authors want to emphasize the importance of this family of proteins in these processes, but in the way it is presented, it does not meet that objective, I do not follow the rationale. On the contrary, it is confusing, it seems as if there were no script.”

Response: We are very sorry that the writing of the introduction in current manuscript does not clearly explain the background of this research. We have adjusted the content of introduction according to your comments.

(1) the expressions “main economic crops”(37) is not correct;

We have changed this sentence according to your opinion: “Soybean (Glycine max) is an economically important food and oilseedcrop planted worldwide.

(2) The way of writing is repetitive and makes it not very fluid throughout the Ms(55-56; 57-58);

Indeed, as you said, the expression in lines 55-58 of the manuscript is unclear, and we have made corrections.

The revised statement is: “The changes in cell wall morphology mediated by expansins are crucial for plant growth, development, and response to external environmental stress.

(3) Seventy-five members of the expansin family have been identified with a total of 75 members” (63-64): this sentence has no sense, and gives an idea that the manuscript is in draft phase;

We apologize for our clerical errors and careless checking. The intended meaning of this sentence is: “In soybean, a total of 75 members of the expansin family have been identified.” We have made corrections in the manuscript.

(4) provide new high-quality genetic resources” (83): what does it mean?

The expression here may be somewhat inaccurate. What we want to express is that the wild soybean GsEXLB14 gene has the potential to improve the tolerance of cultivated soybeans to salt and drought stress, and can be used as a candidate gene for improving the resistance of cultivated soybeans in molecular breeding. In order to better understand, we have changed this sentence to: “This study aims to provide a new candidate wild soybean expansin gene for molecular breeding work related to improving the environmental stress resistance of cultivated soybean.

(5) In this section but also in the discussion: the authors do not indicate to the criteria for classifying extensins into different groups: alpha, beta, gamma... the protein under study belongs to the gamma type, but  it is not clear: each group has been recruited for different responses/functions? Based on? The authors mentioned several examples for the reported role of expansins belonging to different groups in response to different abiotic and biotic stresses in different plant species “jumping” from one to another. I assume that the authors want to emphasize the importance of this family of proteins in these processes, but in the way it is presented, it does not meet that objective, I do not follow the rationale. On the contrary, it is confusing, it seems as if there were no script.”

After carefully reading your comments, we found that the introduction of expansin is indeed not detailed enough. In order to enable readers to better understand the expansin family and the functions it performs, we have reorganized the content related to expansin in the introduction section and added the following content:

Among them, α-expansins rapidly induce creep and stress relaxation of primary cell walls in a pH-dependent manner, carried out in a non-enzymatic manner. β-expansins are usually highly specifically expressed in herbaceous plant pollen. The EXLA and EXLB families are usually smaller in size and are believed to have originated later than the EXPA and EXPB families in terms of system evolution. Although studies on expansin proteins have covered all subfamilies, most research has focused on the EXPA and EXPB families.

The changes in cell wall morphology mediated by expansins are crucial for plant growth, development, and response to external environmental stress. The func-tions performed by the members of four subfamilies are generally similar. At present, many studies have proved that expansins are involved in regulating almost all processes of plant growth and development, such as seed germination, root development, nodule development, leaf growth, stomatal opening and closing, stem elongation, flower development, fruit ripening, seed yield, etc. In addition, expansins are involved in the response process of plants to various abiotic stresses, such as drought, salt, high temperature, low temperature, heavy metals, nutrient deficiency, oxidative stress, etc., and most of them show positive regulatory effects.

  1. Response to comment: “3- Results:

As a general comment: although methodological details should be included in the corresponding materials and methods section, it is essential that the results section includes relevant information on how these results were obtained, the biological material, etc. The results section in the current Ms present data, sample nomenclature, treatments, etc., without explanation of their origin. Furthermore, many of this information does not appear in the materials and methods section either.

“is a stable protein”(91): why?

Figures and table: this is really important, any of the figure legends are self-explanatory. On the contrary, they do not explain the nomenclature used, what the relative expression values refer to, etc. In addition, the format of the figure legends is incorrect. In addition, some sections of the figures are not mentioned in the text. This is the case of part A of figure 3.

There is a lot of relevant information missing, such as that concerning the biological material under study, the treatments, the replicates, some of the procedures…. This information is crucial to follow this work.

“These results suggest that salt and drought stress can induce positive transcription of GsEXLB14 in the roots of wild soybeans.”(106-107): I do not understand: Why “suggest”: the authors shows induction?

(114)The authors have already presented RT PCR data in the previous section, although they do not explain it, but now they mention it. Is it assumed to be end-time PCR? (114). After the epigraph, the paragraph begins without explaining that they are referring to the roots that overexpress the gene under study. As mentioned before, k599, control... what does this nomenclature correspond to?

There is no allusion or explanation that the result of transformation is roots that overexpress the protein under study: is it uniform throughout the root? Is it stable? . genotyping?

In relation to the gene expression results, beyond the legends of the missing figures: reference gene used, replicates, number of times the experiment has been repeated, are the primers specific? Is has been proven that they are not amplifying the endogenous G. max gene? In this sense I have many doubts. The expression values ​​for K 599 are considered to be 1... but what is exactly this sample “control K599”?

(117) “statistical result ”: what does it mean?

(121) “normal growth conditions ”: what does it mean?

(125) is poorly expressed.

(146): Figure 5. Phenotypic quantification results of soybean: is poorly expressed.

(149): “separately”: what does men in this context?

(152): it is assumed that these results come from RNA seq experiment. It is not indicated. Then the validation of some of this genes by q real time : which criteria was used to select those genes from the whole RNS seq data?

There are several incorrect or poorly expressed sentences or expressions: 166,181,191,202, 218, 219…..”

(1) As a general comment: although methodological details should be included in the corresponding materials and methods section, it is essential that the results section includes relevant information on how these results were obtained, the biological material, etc. The results section in the current Ms present data, sample nomenclature, treatments, etc., without explanation of their origin. Furthermore, many of this information does not appear in the materials and methods section either.

Unlike the narrative style of typical manuscript, the journal's writing style is to first present the results, and then clarify the materials and methods. This may cause confusion in understanding the results section. Therefore, we have added some descriptions of materials and methods in the results section to facilitate readers' better understanding. The corresponding changes have been highlighted in the revised manuscript.

(2) is a stable protein”(91): why?

This is the analysis result of GsEXLB14 protein by Expasy online software. We are very sorry for the error in the original manuscript, which has been corrected.

(3) Figures and table: this is really important, any of the figure legends are self-explanatory. On the contrary, they do not explain the nomenclature used, what the relative expression values refer to, etc. In addition, the format of the figure legends is incorrect. In addition, some sections of the figures are not mentioned in the text. This is the case of part A of figure 3.

We apologize for the lack of detailed explanations for the legends and other information in the manuscript. We have added them in the figure captions. In Figure 2, the results show the transcription patterns of GsEXLB14 gene in the roots of wild soybean under salt and simulated drought stress with mannitol. The data were obtained using qRT-PCR, and the vertical axis shows the relative expression levels because a relative quantitative method was used. We have added the following explanation in the figure caption: “The experiment was conducted using a relative quantitative method, and the relative expression level represents the fold in the transcription level of GsEXLB14 gene relative to that at 0 h after treatment.” Similarly, relevant descriptions are also added in Figure 9. In Figure 5, we have added a note about "relative growth" in the caption.

(4) In addition, the format of the figure legends is incorrect.

We have carefully checked the legend in the manuscript and found no inappropriate points. If possible, we would appreciate your help in correcting it.

(5) In addition, some sections of the figures are not mentioned in the text. This is the case of part A of figure 3.

We apologize for the oversight in the description of Figure 3A. Figure 3A shows the schematic diagram of the overexpression GsEXLB14 gene expression vector, which has been added to the Materials and Methods section.

(6) There is a lot of relevant information missing, such as that concerning the biological material under study, the treatments, the replicates, some of the procedures…. This information is crucial to follow this work.

We have explained the information about the source of the plant, cultivation conditions and methods (Original manuscript lines 350-359), experimental repetition (All trials were repeated at least three times, Original manuscript line 455), sample size (Original manuscript lines 117, 145), sample weight (Original manuscript line 358) in the Materials and Methods section of the original manuscript. Regarding the PCR conditions you mentioned, different PCR machines and different brands of DNA polymerases can lead to different PCR conditions. We believe that the PCR conditions are not a key factor for the experiment, and therefore, they are not described in the manuscript. Concerning conventional molecular biology tests, at present, commercial kits are commonly used. We have added the name and brand of the kits used for each test step in the manuscript. Relevant instructions can be easily obtained through the Internet. It would be cumbersome to provide step-by-step instructions. In addition, no special methods were used in this study for molecular biology tests, so detailed instructions were not provided.

(7) These results suggest that salt and drought stress can induce positive transcription of GsEXLB14 in the roots of wild soybeans.”(106-107): I do not understand: Why “suggest”: the authors shows induction?

To avoid any misunderstanding by readers, we have changed this sentence to: “These results showed that salt and drought stress could promote the accumulation of GsEXLB14 gene transcripts in the roots of wild soybean.

(8) (114)The authors have already presented RT PCR data in the previous section, although they do not explain it, but now they mention it. Is it assumed to be end-time PCR? (114). After the epigraph, the paragraph begins without explaining that they are referring to the roots that overexpress the gene under study. As mentioned before, k599, control... what does this nomenclature correspond to?

The results in Figure 2.2 show the transcription changes of the GsEXLB14 gene in the roots of wild soybeans under salt and drought stress. Figure 3B shows the RT-PCR identification results of soybean hairy roots overexpressing the GsEXLB14 gene. These are two different experiments. Wild soybean and soybean are two different species, and the GsEXLB14 gene in this study is derived from wild soybeans, not from soybeans. Unfortunately, we are unable to understand the meaning of this comment. The qRT-PCR used in Figure 2.2 detects the transcription of the GsEXLB14 gene in the roots of wild soybean, demonstrating that the transcription of this gene in the roots of wild soybean is induced by salt and drought stress. Therefore, we cloned this gene from the roots of wild soybean and overexpressed it in the hairy roots of soybean using the Agrobacterium rhizogenes genetic transformation system to test the function of the GsEXLB14 gene on the growth of cultivated soybean roots and their tolerance to salt and drought stress. This is also the core idea of this study. The RT-PCR results in Figure 3B are exactly the detection results of transgenic soybean hairy roots, and have no connection with Figure 2.

(9) There is no allusion or explanation that the result of transformation is roots that overexpress the protein under study: is it uniform throughout the root? Is it stable? . genotyping?

I think it is not necessary to clarify the mechanism of the function of Agrobacterium rhizogenes or Agrobacterium tumefaciens here. Using Agrobacterium for plant genetic transformation is a very common technical means, and the soybean genetic transformation system mediated by Agrobacterium rhizogenes is a very mature system. In this process, the cotyledons of soybean are used as explants, and the resulting hairy roots are induced by Agrobacterium rhizogenes, resulting in high transformation efficiency. The expression vector selected in this study is driven by the strong promoter 35S to overexpress the gene, so the expression in the transgenic root system is uniform and stable.

(10) In relation to the gene expression results, beyond the legends of the missing figures: reference gene used, replicates, number of times the experiment has been repeated, are the primers specific? Is has been proven that they are not amplifying the endogenous G. max gene? In this sense I have many doubts. The expression values ​​for K 599 are considered to be 1... but what is exactly this sample “control K599”?

We have added the description of Figure 3A in the Materials and Methods section according to your suggestion. Regarding Figure 3B, the internal reference gene used in the RT-PCR experiment was "Actin" (Original manuscript line 425), and the number of repeats (Original manuscript line 455) and sample size (Original manuscript line 117) were all described in the original manuscript. Regarding whether the gene was amplified in soybean, the results of Figure 3B showed that the GsEXLB14 gene was not detected in K599-induced hairy roots (empty bacteria, not containing any expression vector), proving the specificity of primer design. We apologize for our oversight in explaining what K599 in Figure 3B refers to in the manuscript. We have added explanations in the figure caption and descriptions in the results and materials methods sections. The soybean hairy roots induced by K599 empty bacteria were used as a non-transgenic control group.

(11) (117) “statistical result ”: what does it mean?

The statistical result is the result of calculation, because the efficiency of transgene should be calculated.

(12) (121) “normal growth conditions ”: what does it mean?

Normal growth conditions refer to the culture medium without NaCl or mannitol, which is the standard medium used as the control group (CK).

(13) (125) is poorly expressed.

We have changed this sentence to: “After 7 days of salt stress treatment, the average number of soybean hairy roots in the K599 control group increased by only 8.96, while the transgenic group increased by 21.34.

(14) (146): Figure 5. Phenotypic quantification results of soybean: is poorly expressed.

We have changed the description of Figure 5 to: “Statistical results of growth indicators of soybean hairy roots overexpressing GsEXLB14 gene under salt and drought stress.”

(15) (149): “separately”: what does men in this context?

This is a modal particle, which can be understood as "one by one". To avoid misunderstandings, it has been deleted.

(16) (152): it is assumed that these results come from RNA seq experiment. It is not indicated. Then the validation of some of this genes by q real time : which criteria was used to select those genes from the whole RNS seq data?

"These data are from RNA sequencing" has been added to the manuscript. The screening criteria for genes in Figure 8 have been described in Materials and Methods 4.6. For Figure 9, the genes selected for qRT-PCR were randomly selected based on Figure 8. Because this is a validation of transcriptome data, we did not set relevant screening criteria, and we added instructions to the Materials and Methods section.

(17) There are several incorrect or poorly expressed sentences or expressions: 166,181,191,202, 218, 219…..

Thank you very much for your careful reading of this manuscript, which has been very helpful in improving our manuscript. We have made changes to the sentences that were incorrectly or unclearly expressed.

  1. Response to comment: “3- Discussion:

The discussion is disorganized and, for all the reasons mentioned, it is difficult to follow throughout it. As mentioned before, in this sections there are as well incorrect or poorly expressed sentences. There is barely discussion of the data.”

Response: In the discussion section of the manuscript, we mainly divided it into two parts. In the first part, we discussed the transcription pattern of GsEXLB14 gene in wild soybean, and listed examples of the transcription patterns of expansin genes in other plants. At the same time, we made an outlook on the specific transcription pattern of GsEXLB14 gene in wild soybean roots and its response to salt and drought stress. In the second part of the discussion, we mainly discussed the potential functions of the GsEXLB14 gene, including the fact that overexpression of the GsEXLB14 gene promotes the growth of hairy roots in cultivated soybean, and analyzed the reasons for this in combination with the hypothesis that the expansion protein performs its function. At the same time, combined with examples of other plants in which expansin proteins improve plant salt and drought tolerance, based on transcriptome data obtained from RNA sequencing, we analyzed the reasons for the overexpression of GsEXLB14 improving the salt and drought tolerance of hairy roots in cultivated soybean from the perspective of changes in the transcription level of genes related to plant resistance regulation.

The amount of data involved in this study is not large, and we believe that the focus may not be on the discussion of data. For example, in terms of changes in gene transcription levels, we focus on trends and whether they are statistically significant. We believe that the highlight of this study is the role of the expansin gene GsEXLB14 obtained in wild soybean in improving the growth and salt and drought tolerance of cultivated soybean. Although the current data cannot support the positive effects of transgenic soybean, the transformation of hairy roots alone is not enough, but as a new discovery, it still has certain significance. Therefore, we did not focus on analyzing gene transcription upregulation folds or transgenic hairy root root length and other related experimental data during the discussion session.

At present, there are many studies on expansin proteins, but most of them focus on the observation of transgenic plant phenotypes or changes in physiological indicators. What is the mechanism behind this is still a relatively vague question. For example, how does expansin protein promote root growth? What genes are affected by overexpression of expansin genes in plants, which affects plant resistance? These questions need to be analyzed in discussions, which are currently lacking in relevant research. Therefore, we focused on analyzing these issues in our discussion and tried to find answers.

  1. Response to comment: “4- Materials and methods: I already comment on that:

A first epigraph that is missing for Plant material, treatments and experimental design and sampling: I consider that is crucial a clear description of the plant material used, plant starting material for each experiment, biological, technical replicates, and indicate each replicate consists in how many samples, material,..?

Many necessary information is missing, such as that concerning the biological material under study, the treatments, the replicates carried out, some of the procedures described, such as PCR conditions, biomolecule extraction protocols, among others.

Concerning the A. rhyzogenes transformation: an explanation about the resulting transgenic material: transient or stable root transformation….It is important to know the nature of the plant material used in  the experiments”

Response: Thank you very much for your suggestion. We have already answered the above questions in our previous response.

Reviewing manuscripts is a pro bono job, so we sincerely respect the suggestions made by each reviewer on our manuscript and sincerely thank you for your hard work in improving the quality of our manuscript. We have made changes to the manuscript based on your comments. If you think that our manuscript still needs further revision or there are still some problems, please feel free to raise them. We look forward to your reply.

Yours sincerely,

Feng Xu

2024-05-28

Reviewer 2 Report

Comments and Suggestions for Authors

In the present study, the authors have investigated the function of GsEXLB14 in soybean. The new insights related to the roles of this gene are well provided. This manuscript has good potential for publishing in PLANTS.

Other comments:

-          Abstract is well provided

-          Line 50: α-expansin is correct. Please edit it

-          Line 60: please provide the gene names in italics and apply in whole text

-          Some changes in the phenotype and molecular responses of transgenic plants can be due to the effect of Agrobacterium rhizogenes, not only the GsEXLB14 gene. How can it be separated?

Author Response

Dear Editor and Reviewer 2:

Thank you for your letter and for the reviewer’ comments concerning our manuscript entitled “Overexpression of wild soybean expansin gene GsEXLB14 en-hanced the tolerance of transgenic soybean hairy roots to salt and drought stresses” (Manuscript ID: plants-3022279). Those comments are all valuable and very helpful for revising and improving our paper, as well as the important guiding significance to our researches. We have studied comments carefully and have made correction which we hope meet with approval. Revised portion are marked in red in the paper. The main corrections in the paper and the responds to the reviewer’s comments are as flowing:

Responds to the reviewer’s comments:

Reviewer #2:

  1. Response to comment: Line 50: α-expansin is correct. Please edit it

Response: Thank you very much for your careful review of our manuscript. Here is the writing error that has been corrected.

  1. Response to comment: Line 60: please provide the gene names in italics and apply in whole text

Response: According to the reviewer 1's comments, the introduction has been adjusted and the sentence has been deleted. We have checked and corrected the gene writing style in the manuscript according to your comment.

  1. Response to comment: Some changes in the phenotype and molecular responses of transgenic plants can be due to the effect of Agrobacterium rhizogenes, not only the GsEXLB14 gene. How can it be separated?

Response: First of all, thank you for your question. This may be a misunderstanding caused by our failure to explain clearly in the original manuscript. In the genetic transformation experiment of soybean hairy roots, the control group used was the hairy roots induced by K599 hairy root-inducing Agrobacterium tumefaciens without any exogenous vectors, thus excluding the influence of Agrobacterium tumefaciens. In order to better understand the design of the experiment, we have added relevant descriptions in the revised manuscript.

We sincerely respect the suggestions made by each reviewer on our manuscript and sincerely thank you for your hard work in improving the quality of our manuscript.

Yours sincerely,

Feng Xu

2024-05-29

Reviewer 3 Report

Comments and Suggestions for Authors

The submitted manuscript to PLANTS_MDPI entitled “Overexpression of wild soybean expansin gene GsEXLB14 enhanced the tolerance of transgenic soybean hairy roots to salt and drought stresses” is interesting to investigate. BUT, following are the comments that need to be addressed before acceptance:

Why did authors study drought and salt stresses together?

”100 mM mannitol-simulated drought” How can be drought stress? It should be oxidative stress. Hence, it should be corrected throughout the manuscript.

Based on this the title should be changed to “Overexpression of wild soybean expansin gene GsEXLB14 enhanced the tolerance of transgenic soybean hairy roots against oxidative stresses”

Why do authors only choose the mentioned specific amounts of two compounds?

The Discussion section is fine. However, the conclusion section needs to be revised extensively.

Author Response

Dear Editor and Reviewer 3:

Thank you for your letter and for the reviewer’ comments concerning our manuscript entitled “Overexpression of wild soybean expansin gene GsEXLB14 en-hanced the tolerance of transgenic soybean hairy roots to salt and drought stresses” (Manuscript ID: plants-3022279). Those comments are all valuable and very helpful for revising and improving our paper, as well as the important guiding significance to our researches. We have studied comments carefully and have made correction which we hope meet with approval. Revised portion are marked in red in the paper. The main corrections in the paper and the responds to the reviewer’s comments are as flowing:

Responds to the reviewer’s comments:

Reviewer #3:

  1. Response to comment: Why did authors study drought and salt stresses together?

Response: This is a misunderstanding caused by our lack of writing skills. The reason why we chose the GsEXLB14 gene and salt and drought stress as the objects of this study is that in a previous study, we found that the GsEXLB14 gene is specifically transcribed in the roots of wild soybean and the transcription levels of this gene is significantly up-regulated under salt and drought stress (Int J Mol Sci. 2022;23(10):5407. doi:10.3390/ijms23105407). The root system is the most direct organ for plants to respond to water and osmotic stress, so the GsEXLB14 gene may perform key functions. In previous work, we failed to conduct in-depth research on this gene, but we chose it as the next research object to carry out this study. To avoid misunderstandings, we have explained it in the revised manuscript.

  1. Response to comment: ”100 mM mannitol-simulated drought” How can be drought stress? It should be oxidative stress. Hence, it should be corrected throughout the manuscript. Based on this the title should be changed to “Overexpression of wild soybean expansin gene GsEXLB14 enhanced the tolerance of transgenic soybean hairy roots against oxidative stresses”

Response: First of all, thank you very much for your advice. When conducting plant-related research, polyethylene glycol or mannitol is often used to simulate drought stress. Therefore, in this study, we chose mannitol to simulate the environment of drought stress. At present, many studies use mannitol to simulate drought stress, such as the recently published paper:

(1) Noman M, Jameel A, Qiang WD, et al. Overexpression of GmCAMTA12 Enhanced Drought Tolerance in Arabidopsis and Soybean. Int J Mol Sci. 2019;20(19):4849.

(2) Saadaoui W, Tarchoun N, Msetra I, et al. Effects of drought stress induced by D-Mannitol on the germination and early seedling growth traits, physiological parameters and phytochemicals content of Tunisian squash (Cucurbita maximaDuch.) landraces. Front Plant Sci. 2023;14:1215394.

(3) Sattar FA, Hamooh BT, Wellman G, et al. Growth and Biochemical Responses of Potato Cultivars under In Vitro Lithium Chloride and Mannitol Simulated Salinity and Drought Stress. Plants (Basel). 2021;10(5):924.

(4) Bai Y, Zhang T, Zheng X, et al. Overexpression of a WRKY transcription factor McWRKY57-like from Mentha canadensis L. enhances drought tolerance in transgenic Arabidopsis. BMC Plant Biol. 2023;23(1):216.

There are many other examples, which are not listed here. Therefore, we believe that using mannitol as a model for drought stress is reasonable.

  1. Response to comment: Why do authors only choose the mentioned specific amounts of two compounds?

Response: For the selection of salt stress concentration, we think that this concentration is reasonable and obtained through pre-experiments for wild soybean and soybean hairy roots. In previous published studies, 150 mM NaCl was also used as the concentration for salt treatment.

(1) Feng X, Feng P, Yu H, et al. GsSnRK1 interplays with transcription factor GsERF7 from wild soybean to regulate soybean stress resistance. Plant Cell Environ. 2020;43(5):1192-1211.

(2) Feng X, Li C, He F, et al. Genome-Wide Identification of Expansin Genes in Wild Soybean (Glycine soja) and Functional Characterization of Expansin B1 (GsEXPB1) in Soybean Hair Root. Int J Mol Sci. 2022;23(10):5407.

Regarding the selection of mannitol concentration, we referred to relevant articles on the application of mannitol in drought stress simulation, and determined the concentration through pre-experiments. Finally, the concentration of mannitol used was determined to be 100 mM. Thank you for your suggestion. We will consider using gradient concentration treatment in subsequent research to better detect the function of this gene.

  1. Response to comment: The Discussion section is fine. However, the conclusion section needs to be revised extensively.

Response: Thank you for your affirmation of the content of our manuscript discussion section. We have revised the conclusion part according to your comments, and the revised content is as follows:

In the current study, we cloned the expansin gene GsEXLB14 from the roots of wild soybean. The transcription levels of GsEXLB14 in the roots of wild soybean significantly increased under salt and mannitol-simulated drought stress. Overexpressing GsEXLB14 gene significantly promoted the growth of cultivated soybean hairy roots and their tolerance to salt and mannitol-simulated drought stress. The data from transcriptome analysis showed that under salt and mannitol-simulated drought stress, the transcripts encoding expansins, peroxidases, H+-ATPases, calcium/calmodulin-dependent protein kinases, and dehydration-response proteins in transgenic soybean hairy roots showed significant accumulation, which may be the reason why transgenic hairy roots grow better under stress.

We sincerely respect the suggestions made by each reviewer on our manuscript and sincerely thank you for your hard work in improving the quality of our manuscript.

Yours sincerely,

Feng Xu

2024-05-29
